# Introgression of chromosomal segments conferring early heading date from wheat diploid progenitor, *Aegilops tauschii* Coss., into Japanese elite wheat cultivars

Shigeo Takumi[1]*, Seito Mitta[2], Shoya Komura[1], Tatsuya M. Ikeda[3], Hitoshi Matsunaka[4], Kazuhiro Sato[5], Kentaro Yoshida[1], Koji Murai[2]*

1 Graduate School of Agricultural Science, Kobe University, Nada, Kobe, Japan, 2 Fukui Prefectural University, Eiheiji, Yoshida, Fukui, Japan, 3 Western Region Agricultural Research Center, National Agriculture and Food Research Organization, Fukuyama, Hiroshima, Japan, 4 Kyushu Okinawa Agricultural Research Center, National Agriculture and Food Research Organization, Chikugo, Fukuoka, Japan, 5 Institute of Plant Science and Resources, Okayama University, Kurashiki, Okayama, Japan

* takumi@kobe-u.ac.jp (ST); murai@fpu.ac.jp (KM)

**Data Availability Statement:** The data sets supporting the results of this article are included

## Abstract

The breeding of agriculturally useful genes from wild crop relatives must take into account recent and future climate change. In Japan, the development of early heading wheat cultivars without the use of any major gene controlling the heading date is desired to avoid overlap of the harvesting time before the rainy season. Here, we backcrossed two early heading lines of a synthetic hexaploid wheat, derived from a crossing between durum wheat and the wild wheat progenitor *Aegilops tauschii*, with four Japanese elite cultivars to develop early heading lines of bread wheat. In total, nine early heading lines that showed a heading date two to eight days earlier than their parental cultivars in field conditions were selected and established from the selfed progenies of the two- or three-times backcrossed populations. The whole appearance and spike shape of the selected early heading lines looked like their parental wheat cultivars. The mature grains of the selected lines had the parental cultivars' characteristics, although the grains exhibited longer and narrower shapes. RNA sequencing-based genotyping was performed to detect single nucleotide polymorphisms between the selected lines and their parental wheat cultivars, which revealed the chromosomal regions transmitted from the parental synthetic wheat to the selected lines. The introgression regions could shorten wheat heading date, and their chromosomal positions were dependent on the backcrossed wheat cultivars. Therefore, early heading synthetic hexaploid wheat is useful for fine-tuning of the heading date through introgression of *Ae. tauschii* chromosomal regions.

## Introduction

Current and future climate change requires more efficient utilization of agriculturally useful genes that are found in the wild relatives of crops [1,2]. Determination of flowering time is

within the paper and its supporting information file. Files containing raw sequence data for the RNA sequencing are available in the sequence read archive of DDBJ (accession number DRA009228).

**Funding:** This work was supported by a Grant-in-Aid from the Ministry of Agriculture, Forestry, and Fisheries of Japan (Development of technologies for mitigation and adaptation to climate change in Agriculture, Forestry and Fisheries, BGW-2202), and by a Grant-in-Aid for Scientific Research (B) No. 16H04862 from the Ministry of Education, Culture, Sports, Science, and Technology of Japan.

**Competing interests:** The authors have declared that no competing interests exist.

one of the most critical traits for environmental adaptation of higher plants to regional growth habitats. In Japan, harvesting common wheat before the rainy season is necessary to avoid severe damage of wheat grains such as pre-harvest sprouting and Fusarium head blight disease. Because Japanese wheat breeders must consider the climate characteristics peculiar to Japan, early heading and flowering time before the rainy season are among the most important traits in Japanese wheat breeding.

Heading/flowering time is largely controlled by major loci, namely *Vrn-1* and *Ppd-1*, in common wheat. The *Vrn-1* loci determine vernalization requirement and are located on the long arms of homoeologous group 5 chromosomes [3], and encode an *APETALA1/FRUIT-FUL*-type MADS-box gene [4]. Structural mutations at the *Vrn-1* loci, including insertions/ deletions in the promoter region and large deletions in the first intron, result in the generation of dominant spring-habit alleles [5,6]. Photoperiodic sensitivity is mainly determined by the *Ppd-1* loci on the short arms of homoeologous group 2 chromosomes [7,8]. A dominant photoperiod-insensitive allele, *Ppd-D1a*, has been generated by a 2-kb deletion in the upstream region of the *Ppd-1* pseudo-response regulator gene [9]. Besides the major loci for vernalization requirement and photoperiodic sensitivity, many minor genes that control wheat heading/flowering time have been detected as quantitative trait loci (QTLs) and assigned to various chromosomes [10–13]. Each of the minor genes, many of which determine narrow-sense earliness, has a small effect on the heading/flowering time, but is available for the fine-tuning of wheat heading and flowering [13,14]. In wheat relatives such as *Triticum monococcum* L. and *Aegilops tauschii* Coss., several QTLs for heading/flowering time have been found [15–18]. The early heading/flowering alleles of these QTLs in wheat relatives would be effective for heading/flowering time control in common wheat.

Common wheat (*Triticum aestivum* L., genome constitution AABBDD) is postulated to be derived from interspecific crossing of cultivated tetraploid wheat (*Triticum turgidum* L., AABB) with the pollen of a wild diploid relative, *Ae. tauschii* (DD) [19,20]. The D-genome donor *Ae. tauschii* is recognized as an effective genetic resource for improving the D genome of common wheat [21,22]. For D-genome improvement, many synthetic wheat hexaploids have been artificially produced through crossing of tetraploid wheat and *Ae. tauschii* [23–25]. The synthetic wheat hexaploids can be used as bridges for introgression of agriculturally useful genes in the *Ae. tauschii* natural population into the wheat D genome [2,21,26]. In our previous study, wide variation in heading date was observed in synthetic hexaploid wheat lines from crossings between a tetraploid wheat cultivar Langdon (Ldn) and diverse *Ae. tauschii* accessions, and some early heading lines were found [25]. The synthetic hexaploid wheat lines share the A and B genomes from Ldn and varying D genomes from *Ae. tauschii*, meaning that the heading time variation is due to the D-genome divergence [27]. Several QTLs for the heading and flowering time variation have been assigned to the D-genome chromosomes in the synthetic hexaploid wheat lines [28]. Early heading/flowering alleles that had never been used in wheat breeding before are expected to be involved in heading/flowering time QTLs on the D-genome chromosomes.

The desirable genes found in *Ae. tauschii* can be introduced into common wheat cultivars through the synthetic wheat hexaploids. However, the large effects of major genes such as *Vrn-1* and *Ppd-1* often mask the detection of small effects of the heading/flowering time QTLs in the mapping populations between the synthetic wheat hexaploids and common wheat cultivars [29]. Therefore, in the present study, we tried to produce early heading wheat lines with the D-genome chromosomal segments derived from *Ae. tauschii* through phenotype-based selection. Then, the introgression segments were presumed in the selected wheat lines using a next-generation sequencing approach. RNA sequencing (RNA-seq) is effective in finding genome-wide polymorphisms anchored to wheat chromosomes [30]. Using RNA-seq-based whole genome

genotyping, we explored the *Ae. tauschii*-derived chromosomal regions conferring the early heading date under the Japanese bread wheat backgrounds.

## Materials and methods

### Plant materials

Two synthetic wheat hexaploids used as parental lines in the present study were derived from interspecific crosses between a durum wheat (*Triticum turgidum* ssp. *durum* (Desf.) Husn.) cultivar Langdon (Ldn) and two accessions, PI476874 and AT47, of *Aegilops tauschii* Coss. [25]. The two synthetic wheat lines with the AABBDD genome, Ldn/PI476874 and Ldn/AT47, were selected as early heading and flowering lines from the 82 synthetic wheat lines with the Ldn-derived AABB genome. Ldn/PI476874 and Ldn/AT47 were crossed with four Japanese elite wheat cultivars, Yukichikara, Kitanokaori, Haruyokoi, and Haruhinode. The $F_1$ plants were 2 or 3 times repeatedly backcrossed with the four Japanese cultivars, and early heading individuals ($BC_2$ or $BC_3$ generations) were selected. The selected $BC_2$ or $BC_3$ individuals were 5 or 6 times selfed to select early heading individuals and to generate homozygous lines. Finally, $BC_2F_6$ or $BC_3F_5$ lines were established as the early heading ones. The selection and fixing were performed in the experimental field of Fukui Prefectural University (36˚43'N, 136˚06'E).

### Measurement of field traits

Seeds of the selected early heading lines and their four parental Japanese wheat cultivars were sown in October 2017, and their agronomic characters were measured in 2018. Plants were grown in the experimental field of Fukui Prefectural University. The plants were space-planted at a 10 cm distance in two rows 20 cm apart.

In total, nine traits were measured: culm length (cm), ear length (cm), number of spikelets per spike, grain number per spike, grain number per spikelet, selfed seed fertility (%), thousand-grain weight (g), liter-grain weight (g), and heading date. Culm length and ear traits were measured using three shoots and the ear of each plant. Fertility (%) was estimated by the seed setting rate of the first and second florets of all spikelets. Two to three plants were examined to measure each trait. The trait averages and standard deviations were calculated, and the statistical significances for difference in means were evaluated using ANOVA.

### Evaluation of grain-related traits

Grain size and shape were measured in each cultivar and selected line using *SmartGrain* software ver. 1.2 [31]. Six parameters for grain size and shape, namely grain area size (AS), perimeter length (PL), grain length (GL), grain width (GW), length-width ratio (LWR), and circularity, were recorded for at least 50 seeds of each cultivar and line according to the *SmartGrain* protocol.

Four grain-related traits, grain hardness, weight, diameter, and moisture, were evaluated using SKCS 4100 (Perten, Stockholm, Sweden). The SKCS hardness index was obtained from crushing a sample of at least 100 kernels from each cultivar and selected line, similar to our previous report [32].

A transverse section of grain was observed by a digital microscope, VHX-900 (Keyence, Osaka, Japan), and a scanning electron microscope (SEM), S-3400N (Hitachi High-Technology, Tokyo, Japan), after the grain was snapped in the middle. SEM observation was conducted without any pretreatment at an accelerating voltage of 8.00 kV under low vacuum conditions of 70 Pa at –25˚C as referred to in our previous reports [32,33].

## PCR analysis of major wheat genes

Total DNA was extracted from leaves of the selected early heading lines and parental accessions. To determine the genotypes of major wheat genes controlling heading date and grain quality-related traits, PCR analysis was conducted using gene-specific primer sets as reported previously [6,34,35, http://www.naro.affrc.go.jp/genome/database/index.html] (S1 Table). PCR conditions were according to the previous reports. A primer set, 5'-CGGGCAAACGGAA TCTACCA-3' and 5'-TGGGGCATCGTGTGGCT-3', was designed for detection of the dominant *Vrn-A1* allele of Ldn, and PCR was performed with a 68°C annealing temperature. PCR products were resolved in agarose gels and visualized under UV light after staining with ethidium bromide.

## RNA-seq-based genotyping

Total RNA was extracted from the seedling leaves of the selected early heading lines and parental accessions including AT47. Paired-end libraries for RNA-seq were constructed from 10 μg of total RNA with a TruSeq RNA Library Preparation kit v2 (Illumina, San Diego, CA, USA) according to the manufacturer's procedure, and then the libraries were sequenced by 300-bp paired-end reads on an Illumina MiSeq sequencer. Five libraries per run were applied for sequencing, and about 5.0 million to 6.8 million reads were obtained per run. The sequenced reads were deposited in the DDBJ Sequence Read Archive under accession number DRA009228. The RNA-seq reads of *Ae. tauschii* PI476874 and Ldn were respectively obtained from the DDBJ Sequence Read Archive accession numbers DRA000536 and DRA007097 [36,37]. The RNA-seq reads of Ldn were obtained using the MiSeq sequencer, whereas the RNA-seq data of PI476874 [36] was derived from the seedling leaf library using a Roche 454 sequencing platform (Roche Diagnostics, Mannheim, Germany).

Quality control of the sequencing reads was performed according to our previous report [30]. The filtered reads were aligned to the reference genome sequence of *T. aestivum* cv. Chinese Spring (CS) version 1 [38] using HISAT2 software version 2.1.0 [39]. Single nucleotide polymorphisms (SNPs) and insertions/deletions (indels) were called with SAMtools [40] and Coval [41] as referred to in our previous report [30]. SNPs and indels were called when the depth of read coverage was $\geq$ 10, and $\geq$ 80% of the aligned reads included nucleotide sequences that differed from sequences of the reference genome sequence. The SNP datasets from the four wheat cultivars were compared with those of the selected early heading lines, and SNPs were called between the wheat cultivars and selected lines. Then the cultivar-selected line SNPs were assigned to the SNP datasets from the two synthetic wheat lines constructed from the SNP data of Ldn and two *Ae. tauschii* accessions, and the synthetic wheat-derived SNPs between the wheat cultivars and selected lines were selected. Chromosomal positions of the SNPs between the wheat cultivars and selected lines and the synthetic wheat-derived SNPs were visualized on the CS chromosomes using the package ggplot2 in the statistical computing environment R [42].

## Results

### Selection and establishment of early heading lines

Two early heading synthetic wheat lines, Ldn/PI476874 and Ldn/AT47, were backcrossed with the four Japanese wheat cultivars, and the early heading plants were selected in the BC$_2$ and BC$_3$ populations for backcrossing with the four cultivars. Then, early heading plants were repeatedly selected from the selfed progenies of the selected BC$_2$ and BC$_3$ plants. Finally, nine homozygous lines were established based on their heading earliness and healthful appearance.

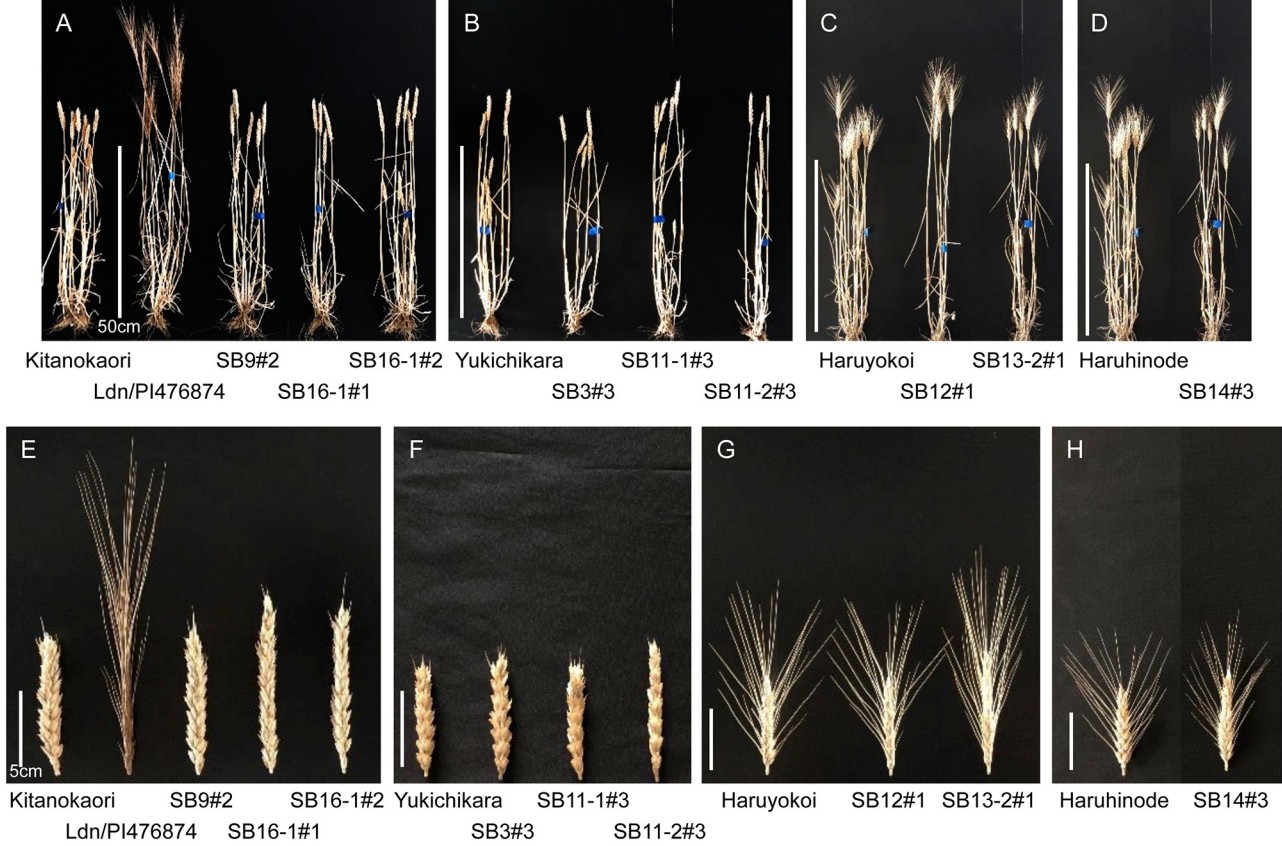

**Fig 1. Photos of plant appearances and spikes in the selected early heading lines and their parental wheat cultivars.**

Three lines, named SB3#3, SB11-1#3, and SB11-2#3, were derived from crossing of Yukichikara and Ldn/AT47. Three lines, SB9#2, SB16-1#1, and SB16-1#2, were from the crosses of Kitanokaori and Ldn/PI476874. Two (SB12#1 and SB13-2#1) and one (SB14#3) lines were respectively established from crossing between Haruyokoi and Ldn/AT47 and between Haruhinode and Ldn/AT47. The whole appearance and spike shape of the selected early heading lines were much more similar to the repeated backcrossed cultivars than to the parental synthetic wheat lines (Fig 1).

## Field performance of the selected early heading lines

Nine field characters of the selected early heading lines were evaluated. In the field of Fukui Prefectural University, the heading date was two to eight days shorter in the selected early heading lines than in the parental cultivars (Table 1). The thousand-grain weight was significantly higher in most selected early heading lines than in the parental cultivars. No significant difference was observed in other field traits. This observation is consistent with the similarity in appearance between the selected early heading lines and the parental cultivars.

## Grain features of the selected early heading lines

Six grain-related traits were measured in the selected early heading lines using *SmartGrain* software. Significant differences in the six traits were observed in most of the selected lines

**Table 1. Plant morphological characters in the selected early flowering lines and their four parental cultivars.**

| Line | Plant height (cm) | Spike length (cm) | No. spikelets per spike | No. grains per spike | No. grains per spikelet | Selfed seed fertility (%) | Thousand-grain weight (g) | Liter-grain weight (g) | Days to heading |
|---|---|---|---|---|---|---|---|---|---|
| Yukichikara | 54 ± 1.9 | 7.2 ± 0 | 15.7 ± 0.3 | 39.3 ± 1.2 | 2.5 ± 0 | 93.6 ± 0.1 | 33.9 ± 0.3 | 679 ± 8.9 | 181 |
| SB3#3 | 49 ± 1.7 | 7.6 ± 0.3 | 15.7 ± 0.3 | 35.0 ± 1.7 | 2.2 ± 0.1 | 89.3 ± 2.2 | 35.4 ± 0.5* | 650 ± 7.1** | 178 |
| SB11-1#3 | 56 ± 1.9 | 7.3 ± 0.2 | 16.0 ± 0 | 37.3 ± 0.9 | 2.3 ± 0.1 | 92.7 ± 2.1 | 37.0 ± 0.3** | 663 ± 4.5 | 178 |
| SB11-2#3 | 54 ± 1.1 | 8.9 ± 0.2** | 16.0 ± 0 | 37.7 ± 1.3 | 2.4 ± 0.1 | 90.6 ± 1.8 | 35.7 ± 0.1** | 679 ± 3.4 | 179 |
| Kitanokaori | 50 ± 0.6 | 8.5 ± 0.4 | 23.3 ± 0.7 | 55.3 ± 5.0 | 2.4 ± 0.2 | 88.4 ± 2.3 | 36.5 ± 0.4 | 694 ± 4.0 | 195 |
| SB9#2 | 53 ± 1.6 | 9.7 ± 0.2 | 22.7 ± 0.3 | 54.3 ± 0.3 | 2.4 ± 0 | 92.6 ± 2.1 | 41.1 ± 0.9** | 673 ± 7.5 | 192 |
| SB16-1#1 | 51 ± 0.7 | 11.0 ± 0.5 | 22.0 ± 0* | 49.7 ± 1.8 | 2.3 ± 0.1 | 92.4 ± 2.7 | 38.8 ± 0.4* | 682 ± 6.3 | 188 |
| SB16-1#2 | 55 ± 0.7* | 7.2 ± 3.0 | 21.7 ± 0.3* | 45.7 ± 2.0* | 2.1 ± 0.1 | 85.4 ± 1.9 | 46.1 ± 1.1** | 665 ± 13 | 189 |
| Haruyokoi | 65 ± 1.2 | 8.4 ± 0.1 | 17.0 ± 0.6 | 35.0 ± 0.6 | 2.1 ± 0.1 | 82.4 ± 1.1 | 39.9 ± 0.3 | 707 ± 4.6 | 195 |
| SB12#1 | 63 ± 1.3 | 8.6 ± 0.3 | 15.3 ± 0.3* | 34.7 ± 0.7 | 2.3 ± 0 | 91.3 ± 1.0* | 45.3 ± 0.5** | 695 ± 7.8 | 188 |
| SB13-2#1 | 70 ± 3.5 | 10.1 ± 0.5** | 17.3 ± 0.7 | 39.3 ± 3.5 | 2.3 ± 0.1 | 89.2 ± 2.9* | 41.4 ± 0.3* | 675 ± 13** | 189 |
| Haruhinode | 56 ± 2.9 | 8.1 ± 0.2 | 16.7 ± 0.7 | 34.0 ± 1.5 | 2.0 ± 0.1 | 86.1 ± 2.3 | 37.6 ± 0.4 | 721 ± 3.6 | 196 |
| SB14#3 | 58 ± 0.8 | 8.0 ± 0.5 | 16.3 ± 0.3 | 28.0 ± 1.5 | 1.7 ± 0.1 | 80.6 ± 1.3 | 48.0 ± 0.5** | 680 ± 4.1** | 191 |

Significant differences between each selected early flowering line and its parental cultivar were indicated by asterisks (*$P < 0.05$; **$P < 0.01$).

(Table 2). Grain length was significantly longer in most of the selected lines compared to the parental cultivars. Length-width ratio and circularity were significantly higher and lower in many selected lines, respectively. These observations indicated that grains of the selected early heading lines exhibited long and narrow shapes compared with those of the parental wheat cultivars.

Wheat grains are generally classified as mealy when the endosperm appears white and floury or as vitreous when translucent and glassy [43]. Vitreousness is related to the mechanical properties of wheat grains [44]. Transverse sections of grains of the parental wheat cultivars were vitreous, whereas those of the synthetic wheat hexaploids were not vitreous (Fig 2). Grains of the selected early heading lines showed a high number of vitreous ones, meaning that they were vitreous like those of the parental cultivars. Sometimes an intermediate

**Table 2. Grain-related traits measured using *SmartGrain* software in the selected early flowering lines and their four parental cultivars.**

| Line | Area size (mm²) | Perimeter length (mm) | Grain length (mm) | Grain width (mm) | Length-width ratio | Circularity |
|---|---|---|---|---|---|---|
| Yukichikara | 16.4 ± 1.6 | 17.3 ± 0.9 | 6.9 ± 0.4 | 3.1 ± 0.2 | 2.2 ± 0.16 | 0.7 ± 0.0 |
| SB3#3 | 14.2 ± 2.5** | 16.3 ± 1.3** | 6.5 ± 0.5** | 2.8 ± 0.3** | 2.3 ± 0.2* | 0.7 ± 0.0* |
| SB11-1#3 | 16.1 ± 2.1 | 17.7 ± 1.0 | 7.1 ± 0.4* | 3.0 ± 0.3* | 2.4 ± 0.3** | 0.7 ± 0.1** |
| SB11-2#3 | 17.5 ± 2.3** | 17.8 ± 1.1** | 7.1 ± 0.5** | 3.2 ± 0.3 | 2.3 ± 0.2 | 0.7 ± 0.0 |
| Kitanokaori | 14.8 ± 2.0 | 16.2 ± 1.1 | 6.4 ± 0.5 | 3.0 ± 0.3 | 2.2 ± 0.2 | 0.7 ± 0.1 |
| SB9#2 | 15.9 ± 2.5* | 17.7 ± 1.3** | 7.2 ± 0.5** | 2.8 ± 0.3** | 2.6 ± 0.3** | 0.6 ± 0.1** |
| SB16-1#1 | 18.6 ± 2.0** | 18.2 ± 0.8** | 7.2 ± 0.3** | 3.3 ± 0.3** | 2.2 ± 0.2 | 0.7 ± 0.0 |
| SB16-1#2 | 18.6 ± 2.4** | 18.4 ± 1.1** | 7.4 ± 0.43** | 3.3 ± 0.3** | 2.3 ± 0.2** | 0.7 ± 0.0 |
| Haruyokoi | 19.0 ± 2.0 | 18.0 ± 0.9 | 7.0 ± 0.4 | 3.5 ± 0.2 | 2.0 ± 0.1 | 0.7 ± 0.0 |
| SB12#1 | 17.3 ± 2.2** | 17.7 ± 1.1 | 7.1 ± 0.5 | 3.1 ± 0.3** | 2.3 ± 0.2** | 0.7 ± 0.1** |
| SB13-2#1 | 17.8 ± 2.3** | 18.3 ± 1.3* | 7.4 ± 0.6** | 3.1 ± 0.2** | 2.4 ± 0.2** | 0.7 ± 0.0** |
| Haruhinode | 16.6 ± 2.0 | 16.5 ± 1.0 | 6.3 ± 0.4 | 3.4 ± 0.2 | 1.8 ± 0.1 | 0.8 ± 0.0 |
| SB14#3 | 19.3 ± 2.6** | 18.6 ± 1.2** | 7.4 ± 0.4** | 3.4 ± 0.3 | 2.2 ± 0.2** | 0.7 ± 0.0** |

Significant differences between each selected early flowering line and its parental cultivar were indicated by asterisks (*$P < 0.05$; **$P < 0.01$).

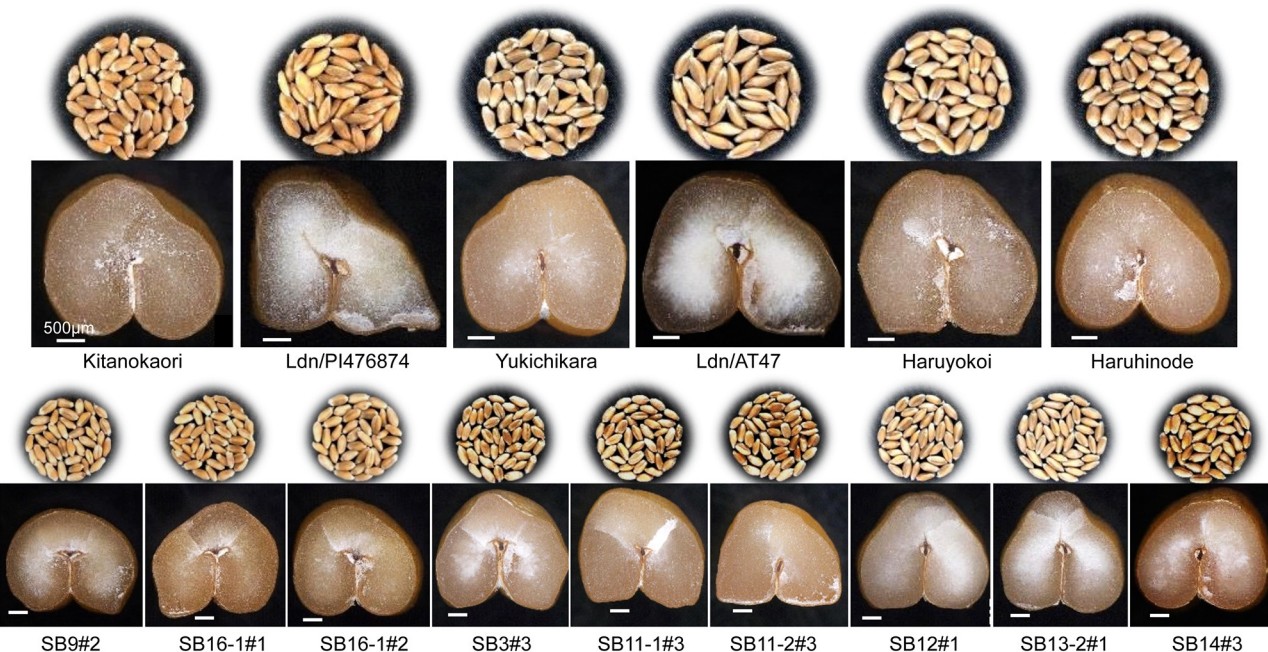

**Fig 2. Photos of mature grains and their transverse sections in the selected early heading lines and their parents.**

phenotype of grain vitreousness appeared in the selected lines, which made it difficult to precisely determine the grain hardness. Because protein content greatly affects vitreousness [45], the grain hardness of the selected lines was then assessed based on genotyping and SEM observation.

The endosperm texture of wheat grains is largely controlled by the *Hardness* (*Ha*) locus on the short arm of chromosome 5D [46]. The *Ha* locus contains two puroindoline protein (PIN)-encoded genes, *Pina* and *Pinb* [47]. The hard kernel texture of common wheat results from a mutation in either *Pina* or *Pinb*, and the normal accumulation of PINA and PINB results in a soft texture [48]. Durum wheat cultivars including Ldn have a very hard texture due to deletion of the *Pina* and *Pinb* genes [49], whereas synthetic wheat hexaploids with the AABBDD genome show a soft kernel texture resulting from transmission of functional *Pina* and *Pinb* alleles from *Ae*. *tauschii* accessions [32,50]. SKCS hardness indexes of the four parental wheat cultivars were high (>63), indicating hard kernel textures (Table 3). The selected early heading lines also exhibited high hardness indexes of the same levels of the parental cultivars. In other traits measured by SKCS, the selected lines showed similar mean values as the parental cultivars.

To compare the internal structures of endosperms in mature grains of the selected and parental lines, the surfaces of transverse sections of the grains were observed using SEM (Fig 3). The surfaces of the starch granules were separated from the matrix proteins and cleaned without any chaff in the parental synthetic wheat lines, whereas the starch granules were embedded in the matrix proteins and broken starch granules were frequently observed in the parental cultivars. Crumbs of the matrix proteins adhered to the starch granule surfaces of the parental cultivars. The phenotypic differences in the grain sections indicated that the endosperm texture of the parental synthetic wheat lines was soft and that of the parental cultivars was hard. The starch granule surfaces of the selected early heading lines were adhered to by the matrix protein crumbs, and the starch granules were embedded in the matrix proteins. The

**Table 3. Grain-related traits measured by SKCS in the selected early flowering lines and their four parental cultivars.**

| Line | Hardness index | Weight (mg) | Diameter (mm) | Moisture (%) |
|------|---------------|-------------|---------------|--------------|
| Yukichikara | 63.8 ± 11.5 | 47.6 ± 5.9 | 3.0 ± 0.2 | 11.1 ± 0.2 |
| SB3#3 | 77.2 ± 13.9 | 36.6 ± 10.0 | 2.7 ± 0.3 | 12.0 ± 0.3 |
| SB11-1#3 | 74.1 ± 13.4 | 39.5 ± 9.7 | 2.8 ± 0.3 | 11.6 ± 0.4 |
| SB11-2#3 | 68.6 ± 12.1 | 46.7 ± 8.1 | 2.9 ± 0.3 | 11.3 ± 0.2 |
| Kitanokaori | 86.8 ± 11.9 | 40.1 ± 5.6 | 2.9 ± 0.2 | 11.8 ± 0.3 |
| SB9#2 | 80.0 ± 10.8 | 41.4 ± 7.4 | 2.8 ± 0.2 | 11.2 ± 0.3 |
| SB16-1#1 | 70.7 ± 12.1 | 54.1 ± 7.0 | 3.1 ± 0.2 | 10.6 ± 0.3 |
| SB16-1#2 | 71.6 ± 12.6 | 55.1 ± 7.5 | 3.1 ± 0.2 | 11.0 ± 0.3 |
| Haruyokoi | 73.7 ± 15.7 | 50.3 ± 7.1 | 3.1 ± 0.3 | 11.5 ± 0.4 |
| SB12#1 | 90.9 ± 12.9 | 42.5 ± 7.8 | 2.8 ± 0.3 | 11.7 ± 0.4 |
| SB13-2#1 | 78.5 ± 12.7 | 45.2 ± 7.8 | 2.8 ± 0.2 | 11.4 ± 0.4 |
| Haruhinode | 88.1 ± 11.2 | 44.5 ± 7.3 | 3.1 ± 0.2 | 11.5 ± 0.3 |
| SB14#3 | 75.0 ± 15.3 | 52.8 ± 9.0 | 3.1 ± 0.3 | 10.7 ± 0.3 |

grain section phenotype of the selected lines was similar to that of the parental cultivars, indicating that the endosperm texture was hard in the selected lines.

## Genotyping of the major wheat genes in the selected early heading lines

To discriminate the allele types of the major agriculturally important genes in the selected early heading lines, PCR analyses were performed using four grain character-related and seven heading time-related gene-specific primer sets. The *Pina-D1a* and *Pinb-D1a* allele combination results in soft-textured wheat grains, whereas the grain texture is changed from soft to hard by providing either *Pina-D1b* or *Pinb-D1b* [51]. The high molecular weight glutenin subunit gene *Glu-D1* and the low molecular weight glutenin subunit gene *Glu-B3* are

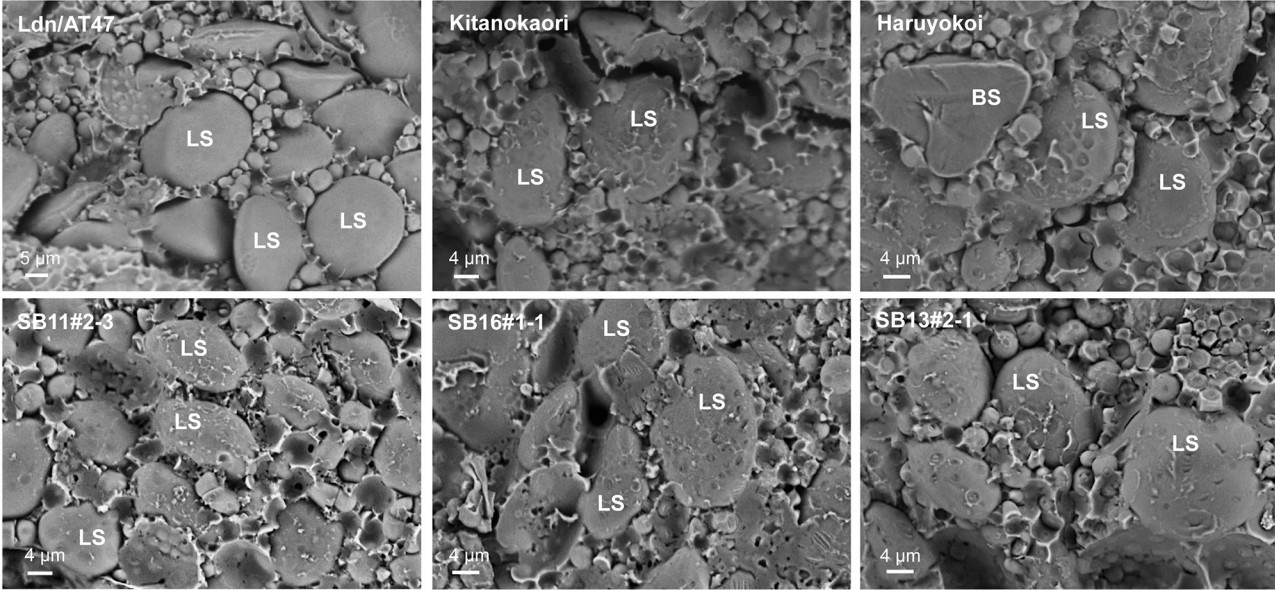

**Fig 3. Scanning electron micrographs of grain sections of the synthetic hexaploid wheat line (Ldn/AT47), two parental bread wheat cultivars (Kitanokaori and Haruyokoi), and three selected early heading lines.** Photos of transverse sections of mature grains are shown for each accession. LS; large starch granule, BS; broken large starch granule.

**Table 4. PCR-based genotyping of the *Pin* and *Glu* alleles in the selected early flowering lines and their four parental cultivars.**

| Line | *Pina-D1* | *Pinb-D1* | *Glu-B3* | *Glu-D1* |
|------|-----------|-----------|----------|----------|
| Synthetics | a | a | l or j | except for d |
| Yukichikara | a | b | b | except for d |
| SB3#3 | a | b | b | except for d |
| SB11-1#3 | a | b | b | except for d |
| SB11-2#3 | a | b | b | except for d |
| Kitanokaori | a | b | j | d |
| SB9#2 | a | b | j | d |
| SB16-1#1 | a | b | j | d |
| SB16-1#2 | a | b | j | d |
| Haruyokoi | b | a | h | d |
| SB12#1 | b | a | h | d |
| SB13-2#1 | b | a | h | d |
| Haruhinode | b | a | h | d |
| SB14#3 | b | a | h | except for d |

important loci for determining wheat grain quality [52,53]. Most of the selected early heading lines provided the same allele combination for the four grain character-related loci as the parental wheat cultivars, but not so for the parental synthetic wheat lines (Table 4). In the two *Vrn-1* loci, *Vrn-B1* and *Vrn-D1*, and three *Ppd-1* homoeologous loci out of the seven heading time-related genes, the selected lines possessed almost the same allele combinations as the parental cultivars (Table 5). Yukichikara and SB3#3 produced no amplified fragment for *Vrn-B1* with the primer sets used in this study. This means that they have novel *Vrn-B1* alleles. The *Vrn-A1* allele of Horuyokoi- and Haruhinode-derived lines corresponded to that of the parental cultivars, whereas the *Vrn-A1* allele was derived from Ldn in the Yukichikara- and Kitanokaori-derived lines.

**Table 5. PCR-based genotyping of the *Vrn-1* and *Ppd-1* alleles in the selected early flowering lines and their four parental cultivars.**

| Line | *Vrn-A1a*[*] | *Vrn-A1*[**] | *Vrn-B1* | *Vrn-D1* | *Ppd-A1* | *Ppd-B1* | *Ppd-D1* |
|------|-----------|----------|----------|----------|----------|----------|----------|
| Synthetics | recessive | dominant | recessive | recessive | b | b | b |
| Yukichikara | recessive | recessive | no band | recessive | b | b | a |
| SB3#3 | recessive | dominant | no band | recessive | b | b | a |
| SB11-1#3 | recessive | dominant | recessive | recessive | b | b | a |
| SB11-2#3 | recessive | dominant | recessive | recessive | b | b | a |
| Kitanokaori | recessive | recessive | recessive | recessive | b | b | a |
| SB9#2 | recessive | dominant | recessive | recessive | b | b | a |
| SB16-1#1 | recessive | dominant | recessive | recessive | b | b | a |
| SB16-1#2 | recessive | dominant | recessive | recessive | b | b | a |
| Haruyokoi | dominant | recessive | dominant | recessive | b | b | b |
| SB12#1 | dominant | recessive | dominant | recessive | b | b | b |
| SB13-2#1 | dominant | recessive | dominant | recessive | b | b | b |
| Haruhinode | dominant | recessive | dominant | recessive | b | b | b |
| SB14#3 | dominant | recessive | dominant | recessive | b | b | b |

[*]The primer set was designed for detection of the *Vrn-A1* alleles of common wheat.

[**]The primer set was designed for detection of the *Vrn-A1* alleles of durum wheat.

## Genome-wide graphical genotyping of the selected early heading lines

After quality control, RNA-seq reads were effectively anchored to the CS chromosomes using the CS reference genome information (IWGSC 2018; Table 6). The RNA-seq reads of two synthetic wheat lines were constructed by the sum of the filtered reads of Ldn and the parental *Ae. tauschii* accession. The PI476874 reads were derived from the Roche 454 platform RNA-seq data [36], and the remaining RNA-seq reads were from the MiSeq 300-bp paired-end sequence data [37 in the present study]. Therefore, the datasets providing high-density SNPs were obtained through comparisons between the two synthetic lines (Ldn/AT47 and Ldn/PI476874) and the four backcrossed wheat cultivars, whereas the SNP numbers were lower in the D-genome chromosomes of the Kitanokaori-Ldn/PI476874 comparison than in the other chromosomes (Fig 4). SNPs called between the backcrossed wheat cultivars and the selected early heading lines were compared with the SNP sets between the two synthetic lines and the four wheat cultivars. Consequently, the alleles derived from the parental synthetic wheat lines were biasedly assigned to the 21 chromosomes of each selected line (Table 7).

The chromosome distribution of the alleles derived from Ldn/AT47 was visualized in the three selected lines of Yukichikara (Fig 5). The three selected lines commonly shared a long region of chromosome 5D from AT47 and a short region of chromosome 5A from Ldn. The chromosome 5A region corresponded to the *Vrn-A1* region examined by PCR-based genotyping (Table 5). The alleles derived from PI476874 were commonly enriched on chromosome 5A, whereas no shared allele from Ldn and PI476874 was detected on the B- and D-genome chromosomes in the three selected lines of Kitanokaori (Fig 6). The two selected lines of Haruyokoi commonly shared a short region on Ldn-derived alleles on the short arm of chromosome 2A from Ldn and a short region on the short arm of chromosome 1D, two regions on the chromosome 2D, and a short region on chromosome 7D from AT47 (Fig 7). A Haruhinode-derived line contained many alleles from Ldn and AT47, and three chromosomal regions with high-density AT47-derived alleles were detected in the selected line on chromosomes 2D, 5D, and 7D (Fig 7). The short arm region of chromosome 2D contained the *Ppd-D1* locus.

**Table 6. Alignments of RNA-seq reads to the CS reference genome and SNP calling.**

| Line | Short reads | Number of filtered short reads | Alignment reads (%) | SNPs |
|---|---|---|---|---|
| PI476874 | 669383 | 348976 | 179414 (51.41%) | 143 |
| AT47 | 4972932 | 3076553 | 2763303 (89.81%) | 58276 |
| Ldn | 6316174 | 4365490 | 3961770 (90.75%) | 40864 |
| Yukichikara | 5704291 | 3235890 | 3006678 (92.92%) | 14984 |
| SB3#3 | 6770706 | 3820073 | 3529836 (92.40%) | 20831 |
| SB11-1#3 | 5968875 | 3498880 | 3240623 (92.62%) | 16888 |
| SB11-2#3 | 5403372 | 3106937 | 2859040 (92.03%) | 17284 |
| Kitanokaori | 5648886 | 3169954 | 2903478 (91.60%) | 15900 |
| SB9#2 | 5387783 | 3340342 | 3081828 (92.26%) | 21751 |
| SB16-1#1 | 6240674 | 3865559 | 3555322 (91.98%) | 20924 |
| SB16-1#2 | 5446262 | 3405431 | 3144968 (92.35%) | 20183 |
| Haruyokoi | 6343349 | 3602895 | 3360936 (93.29%) | 17818 |
| SB12#1 | 6315944 | 3635296 | 3371723 (92.75%) | 19418 |
| SB13-2#1 | 5381735 | 3201166 | 2942918 (91.94%) | 19974 |
| Haruhinode | 5998712 | 3522788 | 3267708 (92.76%) | 15775 |
| SB14#3 | 5123579 | 2921022 | 2709816 (92.77%) | 13753 |

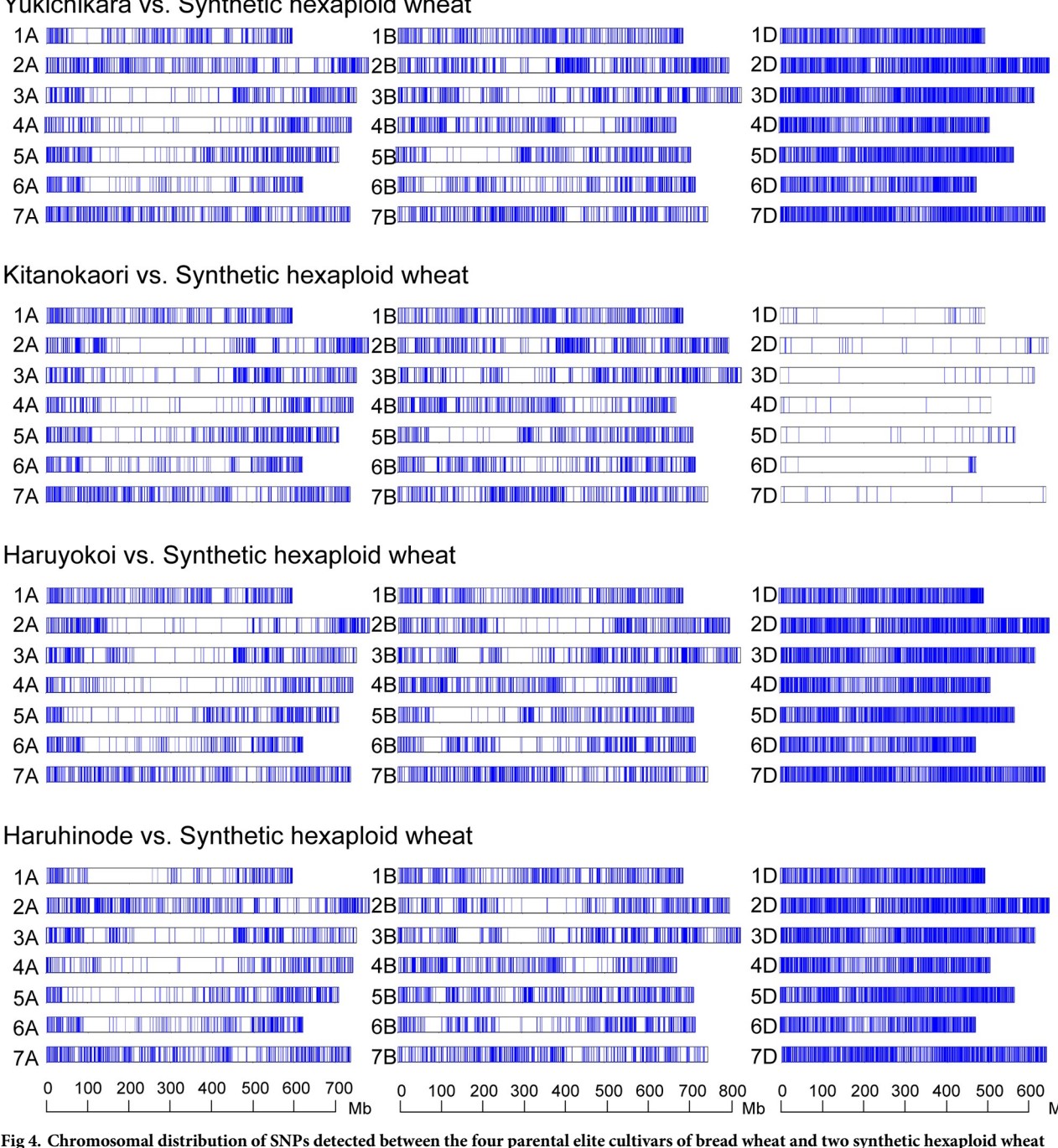

**Fig 4. Chromosomal distribution of SNPs detected between the four parental elite cultivars of bread wheat and two synthetic hexaploid wheat lines.** RNA-seq data of the synthetic hexaploid wheat was generated by combining RNA-seq data of Ldn and of *Ae. tauschii* accessions AT74 or PI476874. The scant SNPs on the D chromosomes in the pair between Kitanokaori vs. the synthetic hexaploid wheat (Ldn/PI476874) were caused by the low coverage data of *Ae. tauschii* PI476874 generated by the Roche 454 sequencing platform.

## Discussion

In total, nine early heading lines were selected and established among progenies from the backcrossing of two synthetic wheat lines with four Japanese elite cultivars of bread wheat.

**Table 7. SNPs between early heading lines and their backcrossed wheat cultivars.**

| Cultivars | Yukichikara | | | Kitanokaori | | | Haruyokoi | | Haruhinode |
|---|---|---|---|---|---|---|---|---|---|
| Line | SB3#3 | SB11-1#3 | SB11-2#3 | SB9#2 | SB16-1#1 | SB16-1#2 | SB12#1 | SB13-2#1 | SB14#3 |
| Total | 5274 | 2976 | 2773 | 2880 | 1489 | 1647 | 1623 | 3682 | 4157 |
| 1A | 1 | 7 | 40 | 1 | 6 | 4 | 1 | 1 | 0 |
| 2A | 108 | 201 | 161 | 1 | 2 | 2 | 339 | 44 | 299 |
| 3A | 340 | 259 | 214 | 1 | 1 | 0 | 95 | 1 | 18 |
| 4A | 673 | 274 | 87 | 0 | 4 | 2 | 1 | 3 | 131 |
| 5A | 52 | 159 | 173 | 637 | 586 | 590 | 258 | 55 | 347 |
| 6A | 0 | 0 | 0 | 9 | 616 | 601 | 1 | 3 | 366 |
| 7A | 25 | 0 | 0 | 120 | 2 | 0 | 3 | 5 | 4 |
| 1B | 947 | 8 | 5 | 1 | 0 | 0 | 7 | 3 | 0 |
| 2B | 12 | 65 | 58 | 0 | 0 | 10 | 131 | 423 | 351 |
| 3B | 30 | 88 | 117 | 6 | 0 | 8 | 3 | 2 | 261 |
| 4B | 134 | 104 | 102 | 0 | 0 | 22 | 0 | 3 | 110 |
| 5B | 8 | 206 | 264 | 42 | 4 | 3 | 0 | 4 | 158 |
| 6B | 1 | 1 | 1 | 162 | 6 | 6 | 1 | 1 | 575 |
| 7B | 346 | 292 | 5 | 1 | 0 | 1 | 0 | 0 | 146 |
| 1D | 0 | 0 | 0 | 7 | 1 | 1 | 104 | 126 | 46 |
| 2D | 0 | 111 | 98 | 3 | 0 | 5 | 484 | 412 | 436 |
| 3D | 34 | 57 | 52 | 0 | 2 | 0 | 1 | 1 | 124 |
| 4D | 1 | 3 | 2 | 0 | 1 | 4 | 1 | 11 | 3 |
| 5D | 2183 | 1138 | 1389 | 1676 | 248 | 219 | 0 | 252 | 422 |
| 6D | 298 | 3 | 4 | 3 | 6 | 2 | 4 | 452 | 15 |
| 7D | 81 | 0 | 1 | 210 | 4 | 167 | 189 | 1880 | 345 |

These selected lines showed a heading date of two to eight days earlier and had significantly increased thousand-grain weight compared to the parental backcrossed cultivars in the field conditions. However, the selected lines phenotypically resembled the parental cultivars and provided the same grain characteristics. The selected early heading lines could be useful for further Japanese bread wheat breeding due to their early heading date and increased thousand-grain weight as derived from the parental synthetic hexaploid wheat lines.

Synthetic hexaploid wheat with the AABBDD genome can enlarge genetic diversity available for common wheat breeding [21,26]. However, many phenotypic differences have been recognized in plant height, spikelet density, grain shape, glume toughness, and so on between synthetic hexaploid wheat and modern wheat cultivars [20,33]. To evaluate the phenotypic effects of the synthetic wheat under the background of bread wheat cultivars, multiple synthetic derivatives have recently been developed in the selfed progeny from the $BC_1$ generation of synthetic wheat backcrossed with a Japanese wheat cultivar [54]. In the present study, we selected the early heading lines from the selfed progeny of $BC_2$ or $BC_3$ plants of the synthetic wheat lines backcrossed with four Japanese elite cultivars of bread wheat. Therefore, their genetic backgrounds were closer to the Japanese cultivars. In fact, no significant difference was observed in many field traits between the selected lines and their parental cultivars (Fig 1, Table 1). Moreover, the mature grains of all selected lines showed high (>60) SKCS hardness indexes (Table 3), and their starch granules tightly adhered to the matrix proteins (Fig 3). These observations indicated that the selected early heading lines exhibited a hard texture. Because the parental synthetic wheat lines clearly showed a soft texture with smoothly rounded starch granules, the hard endosperm texture

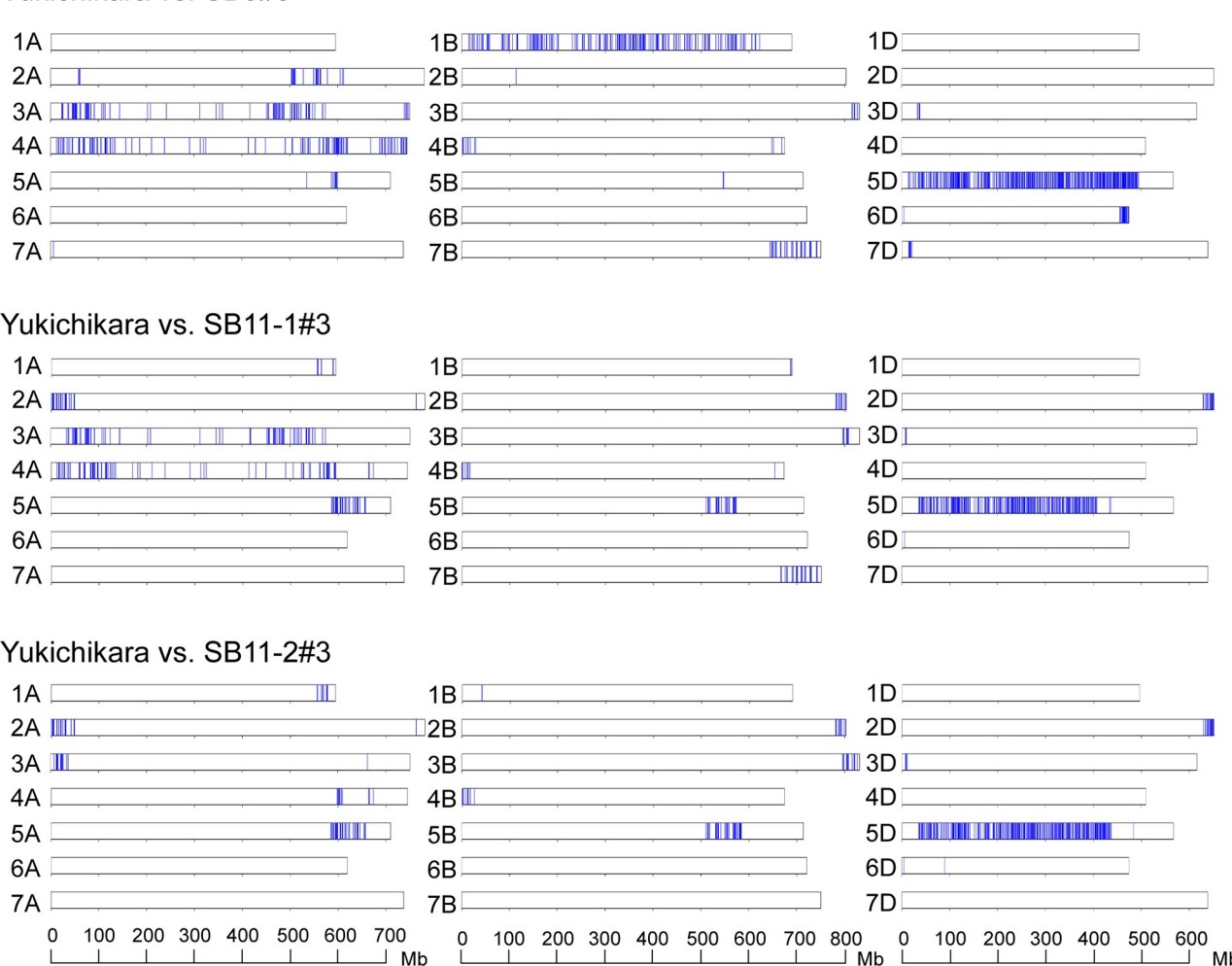

**Fig 5. Genome-wide graphical genotypes of the three selected early heading lines of Yukichikara.** Vertical bars (blue) indicate the SNP sites detected between the three selected lines and their backcrossed parental cultivar Yukichikara on each of the 21 wheat chromosomes.

of the selected lines was transmitted to the selected lines from the parental bread wheat cultivars.

In addition, thousand-grain weight was significantly increased in the selected early heading lines (Table 1). Grain shape was also significantly changed, and the grain length and length-width ratio were increased in most selected lines (Table 2). Round grains are a modern wheat characteristic, while long and narrow grains are a synthetic hexaploid wheat characteristic [33]. It is still unclear whether the increase in thousand-grain weight was due to the long and narrow shape of the grains in the selected early heading lines. A major locus for the grain shape difference between common wheat cultivars and synthetic hexaploid wheat is *tenacious glume 1* on the short arm of chromosome 2D (*Tg-D1*) [33]. *Tg-D1* is located just distally from *Ppd-D1* [20,33,55], and the *Ppd-D1* genotypes of the selected lines of Yukichikara and Kitano-kaori corresponded to that of the parental wheat cultivars (Table 5). Therefore, the genetic locus *Tg-D1* should not be involved in the grain shape difference between the selected early heading lines and their parental wheat cultivars, which was also supported by the free-thresh-ability phenotype of the selected lines.

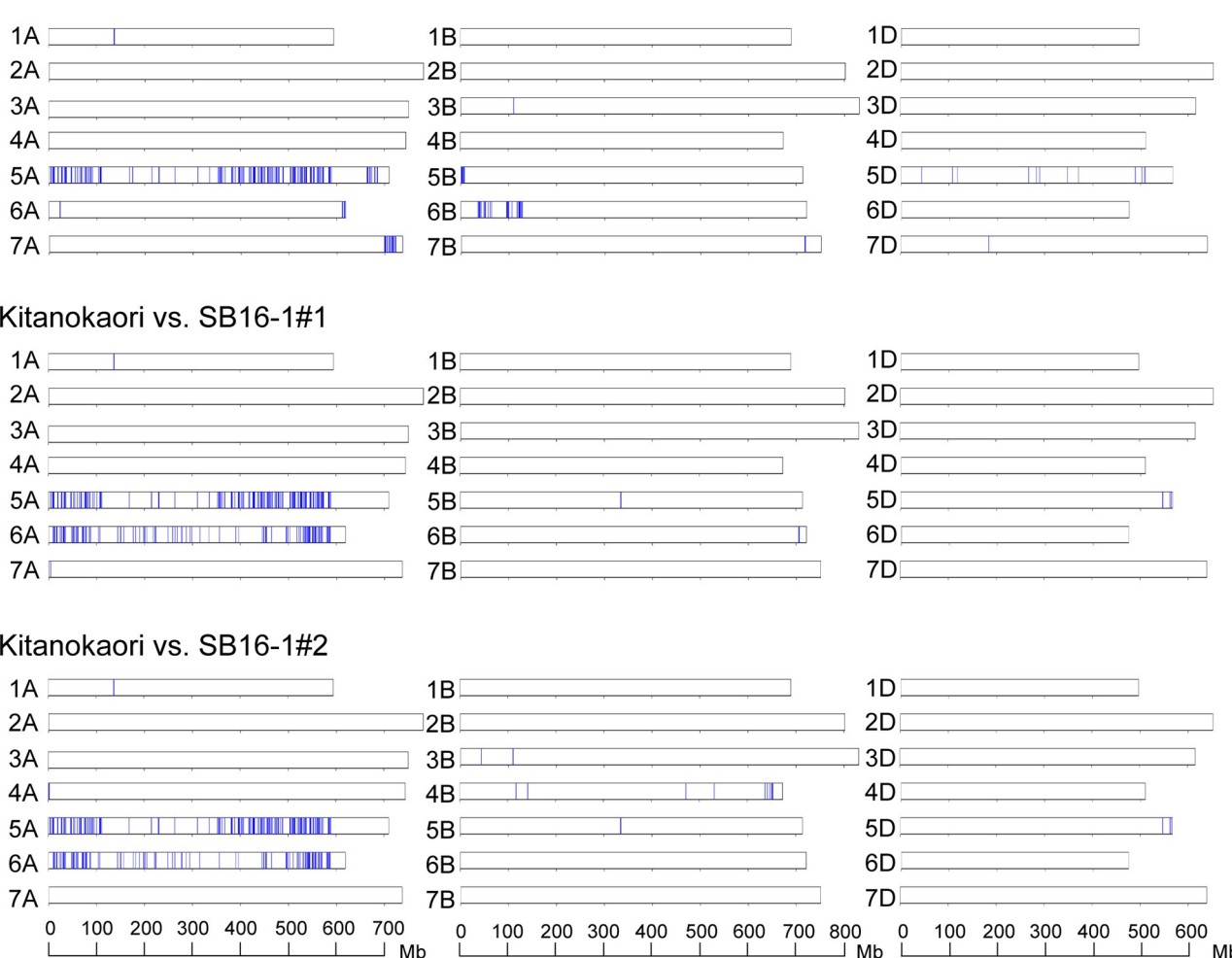

**Fig 6. Genome-wide graphical genotypes of the three selected early heading lines of Kitanokaori.** Vertical bars (blue) indicate the SNP sites detected between the three selected lines and their backcrossed parental cultivar Kitanokaori on each of the 21 wheat chromosomes.

The heading date was two to eight days shorter in the selected early heading lines than in their parental wheat cultivars (Table 1). Genome-wide genotyping using an RNA-seq-based SNP detection approach successfully determined the introgression regions from the parental synthetic hexaploid lines to the selected wheat lines (Figs 5–7). These results strongly suggest that the genetic loci for controlling heading time could be involved in the introgression regions. Actually, the Yukichikara- and Kitanokaori-derived lines shared a chromosomal region on 5A in which the Ldn-derived dominant allele of *Vrn-A1* was present (Table 5). No commonly transmitted region was observed on chromosome 5A in the two Haruyokoi-derived lines, because the backcrossed parent Haruyokoi possessed the *Vrn-A1a* dominant allele. This result indicated that the *Vrn-A1* effect of Haruyokoi on heading earliness was stronger than that of Ldn. Moreover, the length of the introgression regions on 5A was rather different between the Yukichikara- and Kitanokaori-derived lines. The large chromosomal region of chromosome 5A, including the centromeric region, was commonly transmitted from Ldn to Kitanokaori-derived lines and might have had a strong effect on heading earliness

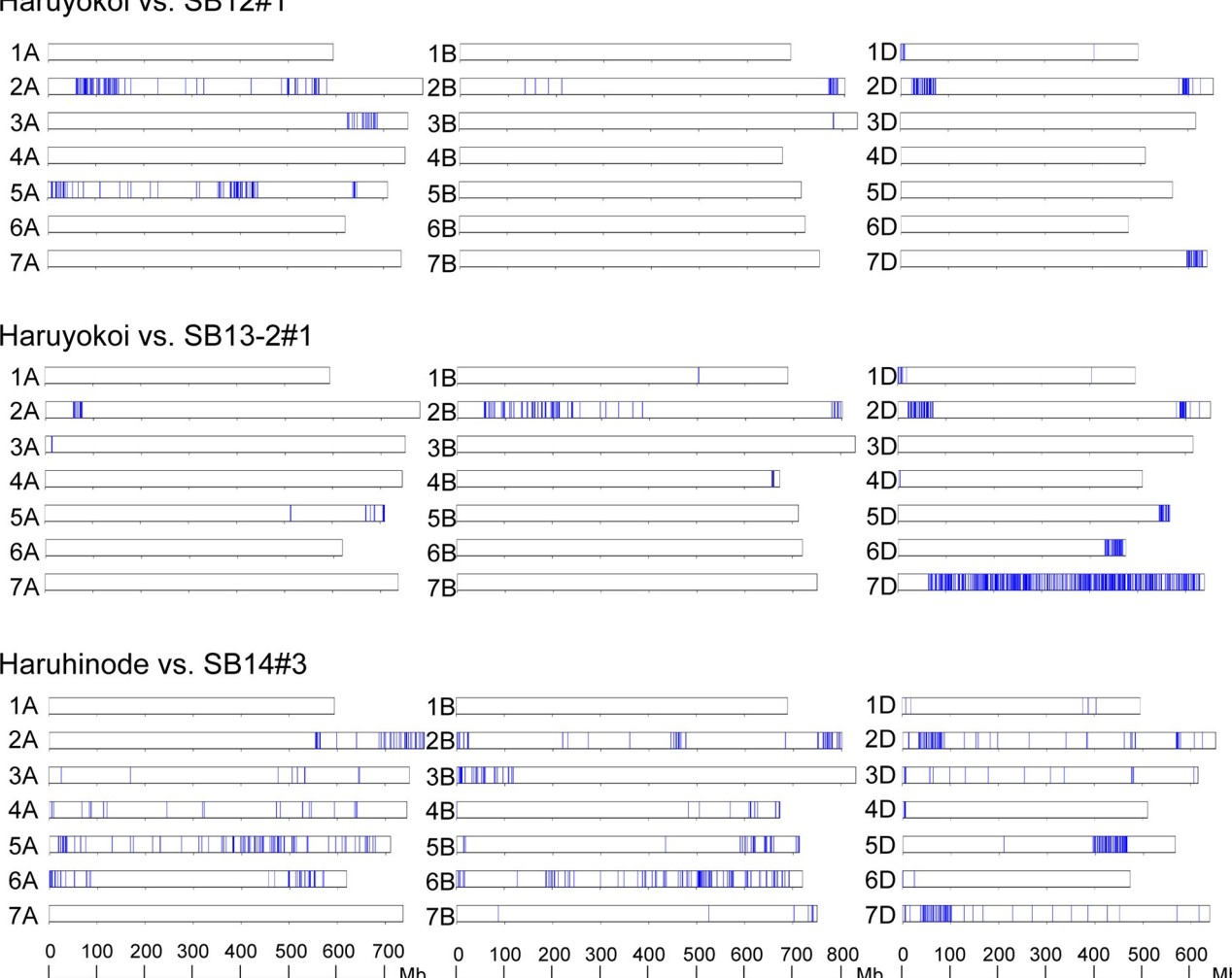

**Fig 7. Genome-wide graphical genotypes of the three selected early heading lines of Haruyokoi and Haruhinode.** Vertical bars (blue) indicate the SNP sites detected between the two selected lines and their backcrossed parental cultivar Haruyokoi and between the selected line and Haruhinode on each of the 21 wheat chromosomes.

under the Kitanokaori background. The length difference of the transmitted 5A regions between the Yukichikara- and Kitanokaori-derived lines could be due to the difference of genetic background. Unlike Kitanokaori-derived lines, the Yukichikara-derived lines commonly shared the long chromosomal fragment including the centromeric region on chromosome 5D (Fig 5). This 5D chromosomal region derived from AT47 could shorten the heading time under the Yukichikara background, but any genetic locus associated with heading time improvement was unclear. The 5D chromosomal region of Yukichikara could be open to improvement, and the 5D region of AT47 could replace that of Yukichikara without any critical damage to field performance.

One short region from Ldn and four short regions from AT47 were commonly shared in the two selected lines of Haruyokoi (Fig 7). The 2D chromosomal region was also detected in the Haruhinode-derived line and contained *Ppd-D1*. Two haplotypes of the *Ae. tauschii Ppd-D1* locus have been recognized to contribute partly to heading time variation [56,57]. The heading date difference between the two *Ppd-D1* haplotypes is quite small and independent of the

photoperiodic response in *Ae. tauschii*. The selected line Haruhinode possessed two additional regions on chromosomes 5D and 7D with high-density AT47-derived alleles. The 7DS region was transmitted from AT47 to one of the two Haruyokoi-derived lines. In addition, the 7D region has been reported as a QTL for heading time in mapping populations of synthetic hexaploid wheat lines and *Ae. tauschii* accessions [18,28]. The 5DL region found in the Haruhinode-derived line has also been reported in the mapping *Ae. tauschii* populations under short-day conditions [18]. These results suggest that the chromosomal regions contain genetic loci controlling heading time with minor effects. The chromosomal regions could shorten heading time under the Haruyokoi and Haruhinode backgrounds, but it is unknown which genes located on the transmitted regions contribute to the early heading date. The effect of the photoperiodic insensitive alleles of *Ppd-1* on heading earliness is quite strong, and such strong effect-providing alleles could hide the effects of the minor QTLs [29]. The minor QTLs for heading time are considered to be narrow-sense earliness genes involved in the fine-tuning of wheat flowering [13,14]. Our application of an RNA-seq-based genotyping approach to the field selection-derived lines successfully detected some commonly shared chromosomal regions. The chromosomal regions detected in the selected early heading lines could contain minor QTLs for heading time and be available for fine-tuning of the heading date in further wheat breeding. Further studies to produce near isogenic lines of each minor QTL and to narrow down the introgression regions should be required to identify the causal genes for these QTLs.

In the present study, we tried to transmit early heading date genes from *Ae. tauschii* to the D-genome chromosomes of Japanese elite cultivars through synthetic hexaploid wheat lines. Phenotype-based selection followed by RNA-seq-based genotyping succeeded in identifying the introgression of *Ae. tauschii* chromosomal regions putatively providing minor QTLs for heading time. In particular, use of the synthetic wheat line Ldn/AT47 promoted heading date changes in multiple Japanese elite cultivars, and the AT47 chromosomal fragments contributed to the heading date earliness. Therefore, *Ae. tauschii* accession AT47 is a useful genetic resource for fine-tuning of heading/flowering dates of Japanese wheat cultivars. However, the chromosomal regions transmitted from the parental synthetic wheat lines to the four wheat cultivars were distinct among the crossing combinations, indicating that the chromosomal positions of the introgression regions appear to be dependent on the genetic background. Marker development based on SNP data enables us to proceed with further marker-assisted selection of early heading alleles in the Japanese wheat breeding processes. Therefore, the field selection approach examined in the present study is effective for the breeding use of minor QTLs together with genome-wide genotyping data.

## Supporting information

**S1 Table. Primer list and PCR conditions for the PCR analyses of wheat major genes.** (PDF)

## Author Contributions

**Conceptualization:** Shigeo Takumi, Koji Murai.

**Data curation:** Shigeo Takumi, Kentaro Yoshida, Koji Murai.

**Formal analysis:** Shigeo Takumi, Shoya Komura, Kentaro Yoshida, Koji Murai.

**Funding acquisition:** Shigeo Takumi.

**Investigation:** Seito Mitta, Shoya Komura, Tatsuya M. Ikeda, Hitoshi Matsunaka, Kazuhiro Sato, Koji Murai.

**Project administration:** Shigeo Takumi.

**Resources:** Shigeo Takumi, Koji Murai.

**Supervision:** Shigeo Takumi, Koji Murai.

**Visualization:** Shigeo Takumi.

**Writing – original draft:** Shigeo Takumi.

**Writing – review & editing:** Tatsuya M. Ikeda, Kazuhiro Sato, Kentaro Yoshida, Koji Murai.

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
