## [Decision Letter · Decision Letter 0]

5 Dec 2019

PONE-D-19-31756

Introgression of chromosomal segments conferring early heading date from wheat diploid progenitor, Aegilops tauschii Coss., into Japanese elite wheat cultivars

PLOS ONE

Dear Dr. Takumi,

Thank you for submitting your manuscript to PLOS ONE. After careful consideration, we feel that it has merit but does not fully meet PLOS ONE’s publication criteria as it currently stands. Therefore, we invite you to submit a revised version of the manuscript that addresses the points raised during the review process.

We would appreciate receiving your revised manuscript by Jan 19 2020 11:59PM. To enhance the reproducibility of your results, we recommend that if applicable you deposit your laboratory protocols in protocols.io, where a protocol can be assigned its own identifier (DOI) such that it can be cited independently in the future. For instructions see: http://journals.plos.org/plosone/s/submission-guidelines#loc-laboratory-protocols

We look forward to receiving your revised manuscript.

Kind regards,

Aimin Zhang, Ph.D.

Academic Editor

PLOS ONE

Journal Requirements:

Reviewers' comments:

Reviewer's Responses to Questions

**Comments to the Author**

1. Is the manuscript technically sound, and do the data support the conclusions?

Reviewer #1: Yes

Reviewer #2: Partly

2. Has the statistical analysis been performed appropriately and rigorously? 

Reviewer #1: Yes

Reviewer #2: Yes

3. Have the authors made all data underlying the findings in their manuscript fully available?

Reviewer #1: Yes

Reviewer #2: Yes

4. Is the manuscript presented in an intelligible fashion and written in standard English?

Reviewer #1: Yes

Reviewer #2: Yes

5. Review Comments to the Author

Reviewer #1: Breeding early heading wheat cultivars could help to advance the harvest time to avoid the rainy season in Japan. Introducing the genes/QTLs in wheat relatives controlling early heading into common wheat is a effective method to improve the heading time. In this study, nine early heading lines possibly harboring the early heading-date genes from Aegilops were developed. Their agricultural traits, kernel traits and quality traits were measured and compared with their parental cultivars. RNA sequencing-based genotyping was also conducted to detect SNP between these nine lines and their parents. The results indicated that the chromosomal regions transmitted from the parental synthetic wheat might shorten the heading date.

However, there still are a few points I would like to see addressed:

1.Please clarify the primers information used in this study.

2.Please unify the font size in Tables and all the manuscript.

3.Please unify the decimal reservation.

Reviewer #2: The authors selected nine early heading wheat lines from the the progeny of of synthetic hexaploid wheat (AABBDD, derived from hybrid of durum wheat with Aegilops tauschii) crossed with four elite cultivars. The lines selected are useful for the improvement of early heading wheat cultivars.

The manuscript was well written and organized. However, the lines used in the study were derived from only two or three times backcross with elite wheat cultivars. It's very difficult to distinguish the genes related to the early heading traits of the derived lines were from Ae. tauschii or the recombination or gene interactions between elite wheat and durum wheat, and it is proved by the fact that no shared sub-genomic D regions detected in those lines from the same cross in this study. More times backcross (>5 times) should be applied to determine D chromosomes regions related to early heading.

By the way, early heading is controlled by QTLs, the chance using major gene controlling the heading date leading to overlap of the harvesting time should be few.

6. PLOS authors have the option to publish the peer review history of their article (what does this mean?). If published, this will include your full peer review and any attached files.

Reviewer #1: No

Reviewer #2: No

---

## [Author Response · Author response to Decision Letter 0]

20 Dec 2019

Dear Editors and reviewers,

Thank you very much for the reviewers’ kind comments and suggestions to our manuscript.

We examined all of the comments, and revised our manuscript according to the comments as followed.

Improved sentences and words were marked by red color in text and tables.

In addition, we improved the format of our manuscript to fit the PLoS ONE’s style requirements.

To Reviewer #1,

Thank you very much for your kind review and polite comments.

1. Please clarify the primers information used in this study.

<response> We made a supplementary table for the primer list and PCR conditions.

2. Please unify the font size in Tables and all the manuscript.

<response> We improve the font size in Tables to 12 point.

3. Please unify the decimal reservation.

<response> We improved Table 2 to unify the decimal reservation.

To Reviewer #2,

Thank you very much for your polite comments.

1. The lines used in the study were derived from only two or three times backcross with elite wheat cultivars. It's very difficult to distinguish the genes related to the early heading traits of the derived lines were from Ae. tauschii or the recombination or gene interactions between elite wheat and durum wheat, and it is proved by the fact that no shared sub-genomic D regions detected in those lines from the same cross in this study. More times backcross (>5 times) should be applied to determine D chromosomes regions related to early heading.

<response> The objective of this research is not production of near-isogenic lines for wheat minor QTLs. As described in Introduction, this research is trial to establishment of early heading lines selected based on their phenotype and to determine the commonly shared chromosomal segments from their parental synthetic wheat hexaploid. We determined the parental synthetic wheat-derived chromosomal regions commonly shared in the selected lines. Based on the results, we discussed relationship between the chromosomal regions and presence of the QTLs for heading date in Discussion by referring to previously published reports. Surely, further researches should be required to clarify whether the identified chromosomal regions in fact control the heading date. Therefore, we added a following sentence in the fifth paragraph of Discussion; ‘Further studies to produce near isogenic lines of each minor QTL and to narrow down the introgression regions should be required to identify the causal genes for these QTLs’.

2. By the way, early heading is controlled by QTLs, the chance using major gene controlling the heading date leading to overlap of the harvesting time should be few.

<response> We agree with the reviewer’s opinion. So, we think that the phenotype-based selection is important to produce early heading lines with some or several days-earliness especially via minor QTLs. Now, we can analyze the whole-genome genotypes using the next generation to identify easily the introgression regions from the parental wild wheat relatives. We think that this point is important in this study as described in Discussion.

We believe that the revised manuscript is now suitable for publication. We look forward to hearing from you at your earliest convenience.

Yours sincerely,

Shigeo Takumi

(Corresponding author)

---

## [Decision Letter · Decision Letter 1]

15 Jan 2020

Introgression of chromosomal segments conferring early heading date from wheat diploid progenitor, Aegilops tauschii Coss., into Japanese elite wheat cultivars

PONE-D-19-31756R1

Dear Dr. Takumi,

We are pleased to inform you that your manuscript has been judged scientifically suitable for publication and will be formally accepted for publication once it complies with all outstanding technical requirements.

With kind regards,

Aimin Zhang, Ph.D.

Academic Editor

PLOS ONE

Additional Editor Comments (optional):

Reviewers' comments:

Reviewer's Responses to Questions

**Comments to the Author**

1. If the authors have adequately addressed your comments raised in a previous round of review and you feel that this manuscript is now acceptable for publication, you may indicate that here to bypass the “Comments to the Author” section, enter your conflict of interest statement in the “Confidential to Editor” section, and submit your "Accept" recommendation.

Reviewer #1: All comments have been addressed

Reviewer #2: All comments have been addressed

2. Is the manuscript technically sound, and do the data support the conclusions?

Reviewer #1: Yes

Reviewer #2: Yes

3. Has the statistical analysis been performed appropriately and rigorously? 

Reviewer #1: Yes

Reviewer #2: Yes

4. Have the authors made all data underlying the findings in their manuscript fully available?

Reviewer #1: Yes

Reviewer #2: Yes

5. Is the manuscript presented in an intelligible fashion and written in standard English?

Reviewer #1: Yes

Reviewer #2: Yes

6. Review Comments to the Author

Reviewer #1: The authors have adequately addressed my comments raised in a previous round of review, I have no addittional comments for the authors. and I feel that this manuscript is now acceptable for publication.

Reviewer #2: The authors has made responses to all my questions and concerns. The manuscript has been carefully reversed. No more comments.

7. PLOS authors have the option to publish the peer review history of their article (what does this mean?). If published, this will include your full peer review and any attached files.

Reviewer #1: Yes: Tao Wang

Reviewer #2: No

---

## [Editor Report · Acceptance letter]

17 Jan 2020

PONE-D-19-31756R1 

Introgression of chromosomal segments conferring early heading date from wheat diploid progenitor, Aegilops tauschii Coss., into Japanese elite wheat cultivars 

Dear Dr. Takumi:

I am pleased to inform you that your manuscript has been deemed suitable for publication in PLOS ONE. Congratulations! Your manuscript is now with our production department. 

With kind regards,

on behalf of

Prof. Aimin Zhang 

Academic Editor

PLOS ONE